# Biological Distribution after Oral Administration of Radioiodine-Labeled Acetaminophen to Estimate Gastrointestinal Absorption Function via OATPs, OATs, and/or MRPs

**DOI:** 10.3390/pharmaceutics15020497

**Published:** 2023-02-02

**Authors:** Kakeru Sato, Asuka Mizutani, Yuka Muranaka, Jianwei Yao, Masato Kobayashi, Kana Yamazaki, Ryuichi Nishii, Kodai Nishi, Takeo Nakanishi, Ikumi Tamai, Keiichi Kawai

**Affiliations:** 1Division of Health Sciences, Graduate School of Medical Sciences, Kanazawa University, 5-11-80 Kodatsuno, Kanazawa 920-0942, Japan; 2Faculty of Health Sciences, Institute of Medical, Pharmaceutical and Health Sciences, Kanazawa University, 5-11-80 Kodatsuno, Kanazawa 920-0942, Japan; 3Department of Molecular Imaging and Theranostics, Institute for Quantum Medical Science, Quantum Life and Medical Science Directorate, National Institutes for Quantum Science and Technology, 4-9-1 Anagawa, Inage 263-8555, Japan; 4Department of Radioisotope Medicine, Atomic Bomb Disease Institute, Nagasaki University, 1-12-4 Sakamoto, Nagasaki 852-8523, Japan; 5Faculty of Pharmacy, Takasaki University of Health and Welfare, 60 Nakaorui-machi, Takasaki 370-0033, Japan; 6Faculty of Pharmaceutical Sciences, Institute of Medical, Pharmaceutical and Health Sciences, Kanazawa University, Kakuma, Kanazawa 920-1192, Japan; 7Biomedical Imaging Research Center, University of Fukui, 23-3 Matsuokashimoaizuki, Eiheiji 910-1193, Japan

**Keywords:** gastrointestinal absorption, anion drugs and medicines, ^125^I-acetaminophen, oral administration, drug transporters

## Abstract

We evaluated the whole-body distribution of orally-administered radioiodine-125 labeled acetaminophen (^125^I-AP) to estimate gastrointestinal absorption of anionic drugs. ^125^I-AP was added to human embryonic kidney (HEK)293 and Flp293 cells expressing human organic anion transporting polypeptide (OATP)1B1/3, OATP2B1, organic anion transporter (OAT)1/2/3, or carnitine/organic cation transporter (OCTN)2, with and without bromosulfalein (OATP and multidrug resistance-associated protein (MRP) inhibitor) and probenecid (OAT and MRP inhibitor). The biological distribution in mice was determined by oral administration of ^125^I-AP with and without bromosulfalein and by intravenous administration of ^125^I-AP. The uptake of ^125^I-AP was significantly higher in HEK293/OATP1B1, OATP1B3, OATP2B1, OAT1, and OAT2 cells than that in mock cells. Bromosulfalein and probenecid inhibited OATP- and OAT-mediated uptake, respectively. Moreover, ^125^I-AP was easily excreted in the urine when administered intravenously. The accumulation of ^125^I-AP was significantly lower in the blood and urinary bladder of mice receiving oral administration of both ^125^I-AP and bromosulfalein than those receiving only ^125^I-AP, but significantly higher in the small intestine due to inhibition of OATPs and/or MRPs. This study indicates that whole-body distribution after oral ^125^I-AP administration can be used to estimate gastrointestinal absorption in the small intestine via OATPs, OATs, and/or MRPs by measuring radioactivity in the urinary bladder.

## 1. Introduction

Many oral drugs and medicines are absorbed into the blood via drug transporters expressed on small intestinal epithelial cells and accumulate in the target tissues and organs. Drug transporters are broadly divided into the solute carrier (SLC) and ATP-binding cassette (ABC) transporters, which contribute significantly to the influx and efflux of drugs [1,2]. In the small intestine, these transporters are expressed on the villous side of the brush border membrane and the vascular side of the basolateral membrane [3] and are mainly involved in the absorption of oral drugs or medicines by the gastrointestinal tract. The SLC transporters on the brush border membrane side (Figure 1) comprise organic anion transporting polypeptide (OATP), mainly OATP2B1 (SLC21A9), and organic cation/carnitine transporter (OCTN), mainly OCTN2 (SLC22A5) and peptide transporter (PEPT)1 (SLC15A1) [4,5,6,7]. In addition, on the brush border membrane side, ABC transporters comprise P-glycoprotein and multidrug resistance-associated protein (MRP)2, whereas, on the basolateral membrane side, these receptors express MRP3 [1,2,5]. Therefore, the expression levels of these transporters play a crucial role in the absorption of oral drugs or medicines. In addition, gastrointestinal diseases that alter the expression levels of these transporters in small intestinal epithelial cells may reduce the gastrointestinal absorption of oral drugs or medicines.

The gastrointestinal absorption function is primarily measured using liquid chromatography/mass spectrometry (LC/MS) [8] or LC/MS/MS. However, this technique is limited because it requires liquid samples (e.g., blood) from the human body after the administration of oral drugs or medicines and cannot accurately reflect the gastrointestinal absorption function for specific SLC transporters on the brush border membrane in the small intestine. Therefore, there is a need for an imaging technique that can accurately estimate gastrointestinal absorption function to personalize drug administration. In our previous study, we used oral administration of iodine-123 labeled m-iodobenzylguanidine (^123^I-MIBG) to estimate gastrointestinal absorption of cationic anticancer drugs or medicines via OCTN and/or OCT in the small intestine by measuring radioactivity in the heart, liver, and urinary bladder [9]. However, this imaging method cannot estimate the absorption function of organic anion drugs and medicines in the gastrointestinal tract.

Acetaminophen (AP, N-acetyl-p-aminophenol, and paracetamol) is an anionic drug that is frequently used as an analog and antipyretic drug. AP is taken up by epithelial cells of the gastrointestinal tract via OATP and OAT [10]. We have previously developed radioiodine-125-labeled AP (^125^I-AP) for the imaging of melanoma and hypothesized that ^125^I-AP might also have affinities for OATP and/or OAT. Therefore, in the present study, we evaluated the whole-body distribution of orally-administered radioiodine-labeled ^125^I-AP to estimate gastrointestinal absorption of anionic drugs or medicines. This imaging technique may help estimate the gastrointestinal absorption of orally administered anionic drugs or medicines, thereby personalizing drug administration. 

## 2. Materials and Methods

### 2.1. ^125^I Labeling of AP and Purification

^125^I-AP was synthesized according to a previously reported protocol [11]. Briefly, ^125^I-AP was synthesized using the chloramine T method by adding 4 mM chloramine T and Na^125^I to 10 mM AP. ^125^I-AP was purified using high-performance liquid chromatography (HPLC, Hitachi, Ibaraki, Japan). Labeling rates were determined using thin-layer chromatography (TLC, silica gel 60 F254; Millipore-Sigma, Burlington, MA, USA). Radiochemical purity was examined with HPLC.

### 2.2. HEK293 and Flp293 Cells Expressing High Levels of Various SLC Transporters

Human embryonic kidney (HEK)293 cells (American Type Culture Collection, Manassas, VA, USA) with high expression levels of OATP1B1 (SLC21A6, NM_006446), OATP1B3 (SLC21A8, NM_019844), OATP2B1 (NM_007256), OAT2 (SLC22A7, NM_006672), and OCTN2 (NM_003060) were used. Flp293 cells derived from HEK293 cells stably express the human α1A-adrenoreceptor, with high expression of OAT1 (SLC22A6, NM_004790) and OAT3 (SLC22A8, NM_004254), were also used. HEK293/OATP1B1, HEK293/OATP1B3, HEK293/OATP2B1, HEK293/OAT2, HEK293/OCTN2, Flp293/OAT1, Flp293/OAT3, and mock control cell lines were established by transfecting HEK293 and Flp293 cells with plasmids encoding each transporter. All cells were cultivated in Dulbecco’s Modified Eagle’s medium (FUJIFILM Wako Chemicals, Osaka, Japan) mixed with 10% fetal bovine serum (Life Technologies, Carlsbad, CA, USA), 1% sodium pyruvate, and 1% penicillin at 37 °C and 5% CO_2_.

### 2.3. Accumulation of ^125^I-AP in HEK293 Cells

Before the experiments with ^125^I-AP, OATP and OAT expressions were confirmed using *p*-^14^C-aminohippuric acid (PerkinElmer Inc., Waltham, MA, USA) in HEK293 and Flp293 cells, and OCTN2 expression was confirmed using methyl-^3^H-4-phenylpyridinium (American Radiolabeled Chemicals Inc., St. Louis, MO, USA) in HEK293 cells [9]. HEK293 cells expressing the respective SLC transporter were seeded at 1.0 × 10^5^ cells/well in 12-well plastic plates in DMEM and cultivated at 37 °C and 5% CO_2_ for 24 h. On the day of the experiments, each cell type was pre-incubated in phosphate-buffered saline (PBS) for 5 min and then incubated with ^125^I-AP (18.5 kBq/well) for 5 min (*n* = 4). After incubation, the cells were washed twice with 600 µL of PBS and lysed with 500 µL of 0.1 M NaOH. Radioactivity was measured using a gamma counter (AccuFLEX γ7000; Hitachi Aloka Medical, Tokyo, Japan). Intracellular protein levels were measured using a bicinchoninic acid protein assay kit (Thermo Fisher Scientific, Waltham, MA, USA), using bovine serum albumin as the standard. 

### 2.4. Accumulation of ^125^I-AP in HEK293 Cells Treated with Inhibitors of Drug Transporters

HEK293 cells expressing the respective SLC transporter were seeded at 1.0 × 10^5^ cells/well in 12-well plastic plates in DMEM and cultivated at 37 °C and 5% CO_2_ for 24 h. The cells were pre-incubated in PBS for 5 min and then incubated with ^125^I-AP (18.5 kBq/well) and 1.0 mM bromosulfalein (Sigma-Aldrich, St. Louis, MO, USA), an inhibitor of OATPs and MRPs [12,13], or 1.0 mM probenecid (Sigma-Aldrich), an inhibitor of OATs and MRPs for 5 min (*n* = 4) [13,14]. After incubation, the cells were washed twice with 600 µL of PBS and lysed with 500 µL of 0.1 M NaOH. Radioactivity was measured using an auto well gamma counter. Intracellular protein levels were measured using a bicinchoninic acid protein assay kit (Thermo Fisher Scientific), using bovine serum albumin as the standard.

### 2.5. Biological Distribution of ^125^I-AP in Normal Mice 

All animal experiments performed in this study were conducted in compliance with the ethical standards of Kanazawa University (Animal Care Committee of Kanazawa University, AP-194046) and the international standards for animal welfare and institutional guidelines. ^125^I-AP was orally and intragastrically administered to ddY mice (male, 8 weeks old; SLC Inc., Hamamatsu, Japan) that had fasted for 6 h, using Fuchigami animal feeding needles (Natsume Seisakusho Co., Ltd., Tokyo, Japan) at a dose of approximately 37 kBq/200 µL per mouse. After 5, 10, 30, and 60 min of administration (*n* = 4), heart blood samples were collected under isoflurane anesthesia (FUJIFILM Wako Pure Chemical Corporation), and mice were sacrificed by cervical dislocation. The organs (thyroid gland, heart, stomach, liver, small intestine, kidney, and urinary bladder) were then quickly collected. The weight of each organ was measured using an electronic balance (AUX220, Shimadzu Corporation, Kyoto, Japan), and radioactivity was measured using an auto gamma counter (AccuFLEX gamma ARC-7010, Hitachi, Ltd., Tokyo, Japan). The results are shown as the percent injected dose (%ID) for the thyroid gland, stomach, small intestine, and urinary bladder and %ID/g wet tissue (%ID/g tissue) for the other organs. As the thyroid gland of mice is quite small and the stomach and small intestine can’t completely remove the contents, it is difficult to measure the accurate weight of these organs. Thus, these organs are indicated by %ID. The urinary bladder results are expressed in %ID because it is undesirable that the radioactivity of urine is corrected by the weight of the urinary bladder. In addition, ^125^I-AP was diluted with saline to approximately 20 kBq/200 µL and administrated intravenously through the tail vein of the mice. After 5, 10, 30, and 60 min of administration (*n* = 4), heart blood samples were collected, and their weights and radioactivity were measured. 

### 2.6. Biological Distribution of ^125^I-AP Administered with an OATP Inhibitor

Mice (ddY, male, 8 weeks old; SLC Inc., Hamamatsu, Japan) fasted for 6 h before using Fuchigami feeding needles for oral and intragastric administration of the drugs. ^125^I-AP was diluted with saline to approximately 20 kBq/100 µL, and 100 µL of ^125^I-AP was orally administered to mice using feeding needles after 5 min of oral administration of bromosulfalein, an inhibitor of OATP. After 5, 10, 30, and 60 min of administration (*n* = 4), heart blood samples were collected under isoflurane anesthesia, and the mice were sacrificed by cervical dislocation. The organs (thyroid gland, heart, stomach, liver, small intestine, kidney, and urinary bladder) were quickly collected. The weight of each organ was measured using an electronic balance, and radioactivity was measured using a gamma counter. The results are shown as %ID or %ID/g of tissue.

### 2.7. Statistical Analysis

*P*-values were analyzed using the two-tailed paired Student’s *t*-test for comparisons between two groups and analysis of variance and the post hoc comparisons with Dunn’s test for comparisons among three groups using GraphPad Prism 8 statistical software (GraphPad Software, Inc., La Jolla, CA, USA). Statistical significance was set as *p* < 0.01.

## 3. Results

### 3.1. ^125^I-AP Accumulation in HEK293 and Flp293 Cells Expressing High Levels of Different SLC Transporters

^125^I-AP was labeled with an efficiency of more than 80% and was obtained with a radiochemical purity of more than 95% after HPLC analysis. Mass-spectrometry data of ^127^I-AP have been reported in our previous study [11]. The date of ^127^I-AP revealed an MH (+) of 278, which matched the calculated MH (+) of 278 C_8_H_8_INO_2_.

The accumulation of ^125^I-AP in HEK293 and Flp293 cells with high expression of an SLC transporter is shown in Figure 2. Before performing the experiments with ^125^I-AP, the function of the SLC transporter in HEK293 and Flp293 cells was confirmed using ^3^H- or ^14^C-labeled substrates [9]. Accumulation of the *p*-^14^C-aminohippuric acid into HEK293 mock cells is 0.19 ± 0.03%ID/mg protein for HEK mock for OATP and OAT2 and into Flp293 mock cells is 0.25 ± 0.06%ID/mg protein for Flp-mock for OAT1 and OAT3. Accumulation of the methyl-^3^H-4-phenylpyridinium into HEK293 mock cells is 0.09 ± 0.06%ID/mg protein for HEK mock for OCTN2. In addition, accumulation of methyl-^3^H-4-phenylpyridinium or *p*-^14^C-aminohippuric acid into HEK293 and Flp293 cells expressing a SLC transporter is as follows: 0.28 ± 0.06%ID/mg protein for HEK293/OATP1B1, 0.27± 0.05%ID/mg protein for HEK293/OATP1B3, 0.34 ± 0.09%ID/mg protein for HEK293/OATP2B1, 0.42 ± 0.11%ID/mg protein for HEK293/OAT2, 0.39 ± 0.08%ID/mg protein for HEK293/OCTN2, 0.46 ± 0.03%ID/mg protein for Flp293/OAT1 and 0.45 ± 0.04%ID/mg protein for Flp293/OAT3.

Compared to HEK293 mock cells, the intracellular accumulation of ^125^I-AP was significantly increased in HEK293/OATP1B1 (*p* < 0.05), HEK293/OATP1B3 (*p* < 0.05), HEK293/OATP2B1 (*p* < 0.01), and HEK293/OAT2 cells (*p* < 0.01). However, the accumulation of ^125^I-AP was not increased in HEK293/OCTN2 cells compared to that in HEK293 mock cells. Compared to Flp-mock cells, the accumulation of ^125^I-AP was significantly increased in Flp293/OAT1 (*p* < 0.05) but not in Flp293/OAT3 cells. In the presence of bromosulfalein, an inhibitor of OATP and MRP, the accumulation of ^125^I-AP was significantly decreased in HEK293/OATP1B1, HEK293/OATP1B3, and HEK293/OATP2B1 cells compared to that in HEK293 mock cells. In the presence of probenecid, an inhibitor of OAT and MRP, the accumulation of ^125^I-AP was significantly decreased in Flp293/OAT1, Flp293/OAT3, and HEK293/OAT2 compared to that in Flp293 or HEK293 mock cells; however, the accumulation of ^125^I-AP in HEK293/OAT2 cells did not decrease to the same level as that in HEK293 mock cells. These results suggest that the uptake of ^125^I-AP was significantly higher in HEK293/OATP1B1, HEK293/OATP1B3, HEK293/OATP2B1, Flp293/OAT1, and HEK293/OAT2 cells than that in mock cells, and that bromosulfalein and probenecid inhibited OATP- and OAT-mediated uptake, respectively.

### 3.2. Biological Distribution of Orally-Administered ^125^I-AP with or without Bromosulfalein and of Intravenously-Administered ^125^I-AP In Vivo

The in vivo stability of intravenously-administered ^125^I-AP has been confirmed in blood and homogenized liver of mice using TLC analysis in our previous study [11]. Although ^125^I-AP was slightly metabolized at 5 min after administration, it remained stable 60 min after administration. The biodistribution of orally-administered ^125^I-AP with or without the oral administration of bromosulfalein and intravenously-administered ^125^I-AP is shown in Table 1. In all the mice, the accumulation of ^125^I-AP in the thyroid gland was low. In the mice group receiving an intravenous administration of ^125^I-AP, ^125^I-AP was easily excreted in the urine. In contrast, the urinary excretion of ^125^I-AP was slower in the oral administration group than in the intravenous administration group. In addition, compared to mice receiving oral administration of ^125^I-AP, those receiving intravenous administration showed a higher accumulation of ^125^I-AP in the blood, heart, small intestine, kidney, and urinary bladder at 5 min after administration. In the group that received oral administration of ^125^I-AP and bromosulfalein, the accumulation of ^125^I-AP in the blood significantly decreased at 5 min after administration, and that in the urinary bladder significantly decreased at 5, 10, and 30 min after administration. However, in the small intestine, the accumulation of ^125^I-AP increased at 5 min after administration. Therefore, biological distribution after oral ^125^I-AP administration is used to estimate gastrointestinal absorption in the small intestine by measuring radioactivity at the early time phase in the urinary bladder.

## 4. Discussion

The gastrointestinal absorption function varies among individuals and in the same individual under disease states. Therefore, an imaging method that can estimate gastrointestinal absorption function would help personalize drug administration. In our previous study, we investigated the distribution of the oral administration of ^123^I-MIBG to estimate the gastrointestinal absorption of cationic anticancer drugs or medicines via OCTN and OCT in the small intestine [9]. This imaging technique may help investigate differences in gastrointestinal absorption, which can affect the activity of cationic anticancer drugs. In the present study, we investigated the biodistribution of orally-administered ^125^I-AP to estimate the absorption function of anionic anticancer drugs or medicines in the gastrointestinal tract.

AP is an anionic drug that is taken up by cells via OATP and OAT [10]. In the present study, ^125^I-AP had affinity for OATP1B1, OATP1B3, OATP2B1, OAT1, and OAT2. Moreover, in the presence of bromosulfalein, an OATP inhibitor, the accumulation of ^125^I-AP in cells with high expressions of OATPs and OAT1 decreased to the same levels as that in HEK293 or Flp293-mock cells (control). However, in the presence of probenecid, an inhibitor of OAT, the accumulation of ^125^I-AP in cells with high OAT2 expression levels did not decrease to the same levels as those in the control cells. These results suggest that the effect of OATPs on the accumulation of ^125^I-AP was more significant than that of OAT2. However, bromosulfalein inhibits not only OATPs but also OAT2 [15]. Therefore, the reduced accumulation of ^125^I-AP in HEK293/OATP1B1, HEK293/OATP1B3, and HEK293/OATP2B1 in the presence of bromosulfalein may also partly involve OAT2. 

Among SLC transporters present on the brush border membrane side of the small intestine, OATP2B1 is the most abundant, whereas OATs, including OAT2, are slightly expressed in the small intestine [4,5,6,7]. In the present study, after intravenous and oral administration of ^125^I-AP in mice, the accumulation of ^125^I-AP in the thyroid gland was low, indicating that ^125^I-AP was little de-iodine in vivo. In addition, intravenously-administrated ^125^I-AP showed high in vivo stability of ^125^I-AP in the blood (94.1 ± 5.1% at 60 min of administration) and liver (88.2 ± 5.5% at 60 min of administration) of mice in our previous study [11]. Therefore, we consider that ^125^I-AP with oral administration also shows high in vivo stability.

Furthermore, compared to control mice, mice receiving oral administration of ^125^I-AP with bromosulfalein showed significantly decreased radioactivity in the blood 5 min after administration but significantly increased radioactivity in the small intestine. These results suggest that OATP2B1 is the key transporter involved in the gastrointestinal absorption of orally-administered ^125^I-AP because the uptake of ^125^I-AP from the brush border membrane to blood was inhibited by the administration of bromosulfalein.

Bromosulfalein also inhibits MRPs that are expressed on both sides of the brush border membrane and basolateral membrane in the small intestine [13]. In addition, because the chemical structure of AP contains a peptide bond in its chemical structure, we evaluated the biological distribution of the oral administration of ^125^I-AP with a peptide substrate, cephradine. The administration of cephradine increased the gastrointestinal absorption of ^125^I-AP (average 9.61, 9.86, 8.96, 2.66 %ID/g in accumulation of blood at 5, 10, 30, 60 min after oral administration with cephradine). Collectively, these results suggest that ^125^I-AP would be absorbed by OATPs and/or OATs on the brush border membrane and by MRP on the basolateral membrane of the small intestine. 

Anionic anticancer drugs or medicines are generally excreted via MRPs of the ABC transporters. Therefore, ^125^I-AP also may have an affinity for MRPs. In the present study, the accumulation of ^125^I-AP in the urinary bladder of mice receiving oral administration of ^125^I-AP and bromosulfalein was significantly decreased within 30 min compared to that in mice orally administered only ^125^I-AP. This effect is due to the inhibition of gastrointestinal absorption by bromosulfalein. Therefore, when gastrointestinal diseases that alter the expression levels of these transporters in small intestinal epithelial cells reduce the gastrointestinal absorption of oral drugs or medicines, it is possible to estimate the gastrointestinal absorption function of the small intestine via OATPs and/or OATs using time–radioactivity curves of the urinary bladder in the early time phase of ^123^I-AP imaging, which was changed from ^125^I to ^123^I. 

Currently, the gastrointestinal absorption function of orally administrated drugs or medicines is primarily measured using LC/MS or LC/MS/MS [8]. We can confirm pharmacokinetics with these data using the samples (blood etc.) after oral administration of a drug or medicine [16]. However, the LC/MS or LC/MS/MS data are difficult to use to evaluate the effect of the individual drug transporter in the small intestine on gastrointestinal absorption. In addition, the absorption function becomes apparent after oral drug or medicine administration. Whole-body imaging after oral ^123^I-AP administration cloud reflects the function of drug transporters prior to oral administration of the drug or medicine. Therefore, it may be possible to control the appropriate dose of the drug or medicine for an individual patient. 

We speculate that the effect of bromosulfalein is limited because we did not observe a reduction in the accumulation of ^125^I-AP in the liver that has high expression levels of OATPs. When ^123^I-MIBG whole-body imaging could estimate the gastrointestinal absorption function via cationic transporters in our previous study, we used DSS-induced experimental colitis mouse as well as the effect of inhibitors to drug transporters in the small intestine because DSS-induced experimental colitis mouse decreased expression levels of OCTN1 [17]. However, there are no disease mouse models which decreased the expression of OATP and/or OAT. If more effective mouse models, such as OATP- or OAT-deficient mouse models, were used, the biological distribution after the oral administration of ^125^I-AP would be slightly different from that reported in the present study. In the future, whole-body imaging after oral ^123^I-AP administration should be performed to estimate gastrointestinal absorption function via OATPs, OATs, and/or MRPs in the small intestine.

## 5. Conclusions

The present study demonstrates that whole-body distribution after oral ^125^I-AP administration can be used to estimate gastrointestinal absorption of anionic drugs or medicines in the small intestine via OATPs, OATs, and/or MRPs by measuring radioactivity in the urinary bladder. Before oral administration with a drug or medicine, whole-body imaging after oral ^123^I-AP administration could estimate the function of OATPs, OATs, and/or MRPs in the small intestine and lead to control of the appropriate dose of orally administrated drugs or medicines for an individual patient as personalized medicine.

## Figures and Tables

**Figure 1 pharmaceutics-15-00497-f001:**
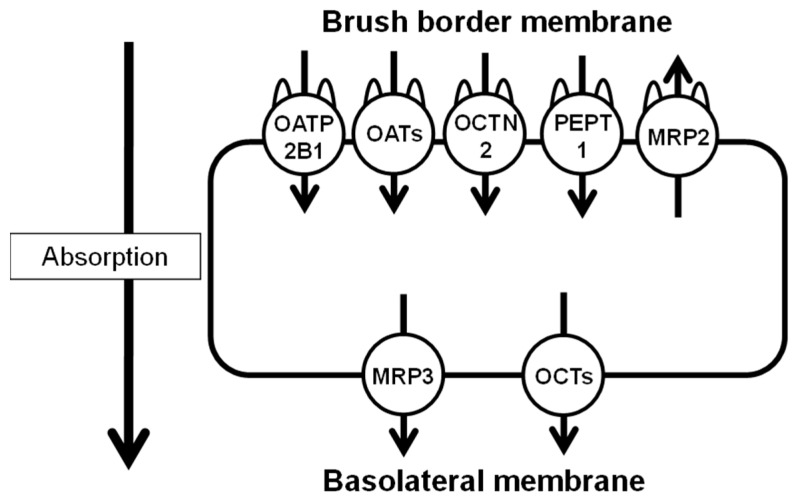
Solute carrier and ATP-binding cassette transporters are mainly expressed on small intestinal epithelial cell membranes. OATP, organic anion transporting polypeptide; OCTN, organic cation/carnitine transporter; PEPT, peptide transporter; MRP, multidrug resistance-associated protein.

**Figure 2 pharmaceutics-15-00497-f002:**
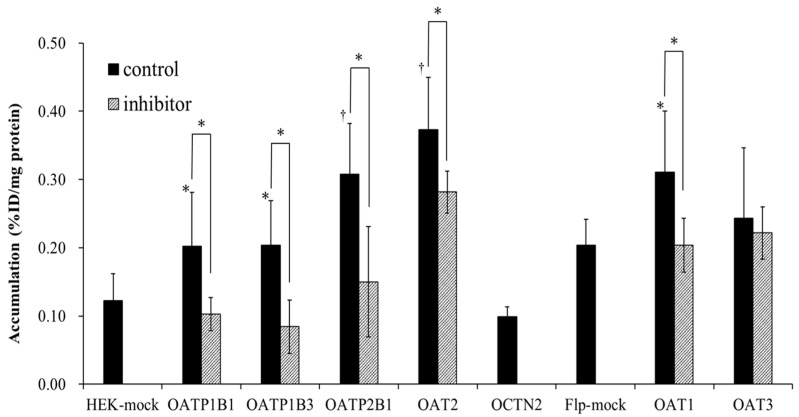
Accumulation of ^125^I-AP in HEK293 and Flp293 cells with high expression of a solute carrier transporter (*n* = 4). Before the experiments with ^125^I-AP, the function of the SLC transporter in HEK293 and Flp293 cells was confirmed using ^3^H- or ^14^C-labeled substrates. Accumulation of the *p*-^14^C-aminohippuric acid into HEK293 mock cells is 0.19 ± 0.03%ID/mg protein for HEK mock for OATP and OAT2 and into Flp293 mock cells is 0.25 ± 0.06%ID/mg protein for Flp-mock for OAT1 and OAT3. Accumulation of the methyl-^3^H-4-phenylpyridinium into HEK293 mock cells is 0.09 ± 0.06%ID/mg protein for HEK mock for OCTN2. In addition, accumulation of methyl-^3^H-4-phenylpyridinium or *p*-^14^C-aminohippuric acid into HEK293 and Flp293 cells expressing a SLC transporter is as follows: 0.28 ± 0.06%ID/mg protein for HEK293/OATP1B1, 0.27± 0.05%ID/mg protein for HEK293/OATP1B3, 0.34 ± 0.09%ID/mg protein for HEK293/OATP2B1, 0.42 ± 0.11%ID/mg protein for HEK293/OAT2, 0.39 ± 0.08%ID/mg protein for HEK293/OCTN2, 0.46 ± 0.03%ID/mg protein for Flp293/OAT1 and 0.45 ± 0.04%ID/mg protein for Flp293/OAT3. ^125^I-AP accumulation is significantly increased in HEK293/OATP1B1, HEK293/OATP1B3, HEK293/OATP2B1, HEK293/OAT2, and Flp293/OAT1 cells compared to that in mock cells. The addition of bromosulfalein, an inhibitor of OATP and MRP, inhibits ^125^I-AP uptake into HEK293/OATP1B1, HEK293/OATP1B3, HEK293/OATP2B1 cells. The addition of probenecid, an inhibitor of OAT and MRP, inhibits ^125^I-AP uptake into HEK293/OAT2 and Flp293/OAT1 cells. ^†^
*p* < 0.01 and * *p* < 0.05 vs. mock cells and cells loaded with both inhibitors. OATP, organic anion transporting polypeptide; OCTN, organic cation/carnitine transporter; PEPT, peptide transporter; MRP, multidrug resistance-associated protein.

**Table 1 pharmaceutics-15-00497-t001:** Biological distribution of ^125^I-AP in mice by oral administration, with or without bromosulfalein, and by intravenous administration.

Time after ^125^I-AP Administration					
Mice	Organ	5 min	10 min	30 min	60 min
Oral administration	Blood	3.97 ± 0.58	4.53 ± 0.71	2.76 ± 0.68	1.19 ± 0.30
	Thyroid gland	0.22 ± 0.08	0.04 ± 0.00	0.09 ± 0.02	0.13 ± 0.07
	Heart	1.48 ± 0.46	1.59 ± 0.16	1.15 ± 0.33	0.44 ± 0.15
	Stomach	62.7± 9.91	38.5 ± 7.17	22.9 ± 9.78	15.9 ± 8.55
	Liver	2.41 ± 0.82	2.38 ± 0.75	1.42 ± 0.27	0.54 ± 0.13
	Small intestine	4.39 ± 2.48	15.6 ± 5.89	16.6 ± 5.76	19.6 ± 7.41
	Kidney	4.11 ± 0.25	6.86 ± 2.98	3.53 ± 0.39	5.78 ± 4.38
	Urinary bladder	0.23 ± 0.13	2.22 ± 12.5	17.8 ± 7.29	16.6 ± 10.8
Oral administration with bromosulfalein	Blood	1.10 ± 0.33 *	3.38 ± 1.76	2.40 ± 0.78	1.11 ± 0.11
	Thyroid gland	0.18 ± 0.09	0.07 ± 0.03	0.08 ± 0.01	0.13 ± 0.02
	Heart	1.53 ± 0.63	1.25 ± 0.68	0.85 ± 0.28	0.43 ± 0.06
	Stomach	57.0 ± 35.1	56.3 ± 23.2	24.8 ± 15.0	11.3 ± 1.79
	Liver	1.81 ± 0.30	1.75 ± 0.68	1.22 ± 0.31	0.64 ± 0.18
	Small intestine	15.2 ± 6.85 *	16.4 ± 2.79	19.0 ± 4.68	18.4 ± 5.79
	Kidney	4.22 ± 1.44	5.50 ± 2.06	5.98 ± 2.42	2.01 ± 0.71
	Urinary bladder	0.06 ± 0.02 *	0.20 ± 0.23 *	7.20 ± 2.71 *	17.8 ± 0.59
Intravenous administration	Blood	7.95 ± 2.15 ^†^	3.51 ± 1.13	1.55 ± 0.43	0.79 ± 0.10 *
	Thyroid gland	0.05 ± 0.01 *	0.05 ± 0.04	0.06 ± 0.02	0.10 ± 0.05
	Heart	2.99 ± 0.60 *	1.14 ± 0.17	0.49 ± 0.12 *	0.26 ± 0.06 *
	Stomach	1.04 ± 0.57 ^†^	1.27 ± 0.30 ^†^	1.01 ± 0.44 †	1.05 ± 0.41 *
	Liver	2.81 ± 0.19	1.07 ± 0.08 ^†^	0.56 ± 0.15 *	0.32 ± 0.04 *
	Small intestine	8.52 ± 0.99	8.31 ± 0.88	8.56 ± 1.56 *	9.21 ± 0.67 ^†^
	Kidney	19.4 ± 3.13 *	12.1 ± 7.89	6.85 ± 8.23	4.30 ± 2.84
	Urinary bladder	10.60 ± 7.65 *	16.6 ± 19.0	23.7 ± 22.0	42.7 ± 16.7

%ID indicates the percent injected dose for the thyroid gland, stomach, small intestine, and urinary bladder. %ID/g indicates the percent injected dose per gram of blood, heart, liver, and kidney. Values represent the mean ± standard deviation of four mice. * *p* < 0.01 and ^†^
*p* < 0.05 compared to oral administration.

## Data Availability

The authors confirm that the data supporting the findings of this study are available within the article.

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
