# Peer review of "Biological Distribution after Oral Administration of Radioiodine-Labeled Acetaminophen to Estimate Gastrointestinal Absorption Function via OATPs, OATs, and/or MRPs"

_pharmaceutics, 2023, doi:10.3390/pharmaceutics15020497_

Round 1

Reviewer 1 Report

Authors evaluated the biological distribution of radioiodine-labeled acetaminophen and estimated GI absorption of I-AP via OATPs, OATs and/or MRPs. Overall, aim of the study seems interesting; however, I suggest following recommendations to further improve the manuscript; 

1. Introduction: Authors mentioned that LC/MS method is usually used to quantify drug absorption in blood and describes that it has limitation of liquid samples. However, LC-MS and LC-MS/MS can be used to quantify drug content in blood as well as in different organs by using validated extraction method. Authors should confirm their statement.

2.  I feel the control is missing in the study for validation of results. If possible, authors should use non labelled AP as control and quantify the drug using other validated/reported approach. Then, compare the result with I-AP with imaging method. Otherwise, authors can explain the reason of their experimental design choice. Non labelled AP and I-AP will follow same path?

3. Only one anionic drug I-AP is used for the study. How can authors explain that the approach is useful for other drugs as well. 

4. Authors follow a reasonable experimental design; however, I suggest the comparison with other techniques is missing. I suggest to cite some references in discussion to compare their results with other approaches.

5. Conclusion should be extended and summarize main finding and future prospect.

Author Response

Thank you for your comments. We can reply to your comments as below.

  1. Introduction: Authors mentioned that LC/MS method is usually used to quantify drug absorption in blood and describes that it has limitation of liquid samples. However, LC-MS and LC-MS/MS can be used to quantify drug content in blood as well as in different organs by using validated extraction method. Authors should confirm their statement.

Answer: Thank you for your comments regarding the LC/MS and LS/MS/MS. Our goal is to estimate the function of gastrointestinal absorption via the SLC and/or ABC transporters in the small intestine. However, even if we can quantify the drug content in liquid sample (blood etc.), it does not reflect only gastrointestinal absorption via SLC and/ABC transporters in the small intestine. Therefore, we considered to use nuclear medicine imaging method with 123I-AP that could reflect the function of these transporters.

  1. I feel the control is missing in the study for validation of results. If possible, authors should use non labelled AP as control and quantify the drug using other validated/reported approach. Then, compare the result with I-AP with imaging method. Otherwise, authors can explain the reason of their experimental design choice. Non labelled AP and I-AP will follow same path?

Answer: This study was designed to estimate the gastrointestinal absorption of anionic drugs from the biological distribution of orally administered 125I-AP, not to confirm the kinetics of non labelled AP. Therefore, it is not necessary that AP and 125I-AP follow in the same way.

  1. Only one anionic drug I-AP is used for the study. How can authors explain that the approach is useful for other drugs as well.

Answer: This study was designed to estimate the gastrointestinal absorption of anionic drugs from the biological distribution of orally administered 125I-AP. In this study, it is important that 125I-AP can reflect the function of OATP, OAT and MRP transporters in small intestine because many anionic drugs are absorbed via these transporters. We have already reported that 123I-MIBG can estimate the gastrointestinal absorption of cationic drugs from the biological distribution. Therefore, combining 123I-AP, which changes 125I to 123I, and 123I-MIBG can cover estimation of the gastrointestinal absorption of anionic and cationic drugs.

  1. Authors follow a reasonable experimental design; however, I suggest the comparison with other techniques is missing. I suggest to cite some references in discussion to compare their results with other approaches. 

Answer: We have described a comparison with LC/MS and LC/MS/MS. Currently, these methods have been used estimate gastrointestinal absorption function. Although liquid sample (blood etc.) collected over time after oral administration with drugs and medicines can reveal pharmacokinetics and absorption function in the small intestine, they can’t reflect only the function of specific transporters in the small intestine. Whole-body imaging after oral 123I-AP administration cloud estimate drug transporter function in the small intestine by measuring radioactivity in urinary bladder and control the appropriate dose of drugs and medicines for an individual patient.

  1. Conclusion should be extended and summarize main finding and future prospect.

Answer: We added a future prospect in the conclusion. Before oral administration with a drug or medicine, whole-body imaging after oral 123I-AP administration could estimate the function of OATPs, OATs and/or MRPs in the small intestine and lead to control the appropriate dose of orally administrated drugs for an individual patient as personalized medicine.

Reviewer 2 Report

This article developed a potential method that can be used to estimate gastrointestinal absorption of anionic drugs. This study indicates that whole-body  distribution after oral 125 I-AP administration can be used to estimate gastrointestinal absorption in  the small intestine via OATPs, OATs, and/or MRPs by measuring radioactivity in the urinary bladder .

In addition , some issue concerned:

1.In the 2.1 section, 125I-AP was synthesized by the author’s laboratory, and can the author provide the indentified result of  MS data as supporting imformation? When 125I-AP was purified, make sure which type of  HPLC was used ( line 95) . And make sure whether radiochemical purity was determined by TLC in line 96 ,which is contradicted with HPLC analysis in line 169.

2.Various SLC Transporters was high levels by transfecting HEK293 and Flp293 cells with plasmids encoding each transporter ,how did the author conclude that ,could the author provide the RT-PCR and western blotting results?

3.Data of Fig.2 are presented as mean and standard deviation (SD),please provide the sample number (n=?)

4.Why  did the results of thyroid gland, stomach, small intestine and urinary  bladder  show  as % ID ,the other organs (blood, heart, liver and kidney) show as  % ID/g?

5.Please supply  the procedure  after inhibitor incubated in section 2.4,such as inhibitor incubated time, detected time, washed procedure , protein extracting procedure and so on.

6.Please supply analysis of time-activity curves for important organs.

7.Why did the author not provide the experiment data about the disease or related transporters deficient mouse models after the oral administration of 125I-AP?

Author Response

Thank you for your comments. We can reply to your comments as below.

1.In the 2.1 section, 125I-AP was synthesized by the author’s laboratory, and can the author provide the indentified result of MS data as supporting imformation? When 125I-AP was purified, make sure which type of HPLC was used (line 95). And make sure whether radiochemical purity was determined by TLC in line 96, which is contradicted with HPLC analysis in line 169.

Answer: Thank you for pointing this out. The labeling rate of 125I-AP was determined using TLC and the radiochemical purity was examined by HPLC. In addition, mass spectrometry date of 127I-AP had revealed MH (+) of 278, which matched the calculated MH (+) of 278 C8H8INO2 in our previous study (Zhu, W.J. et al. Nucl Med Biol. 2018, 59, 16–21.). We added these sentences in the materials and methods and results.

2.Various SLC Transporters was high levels by transfecting HEK293 and Flp293 cells with plasmids encoding each transporter, how did the author conclude that, could the author provide the RT-PCR and western blotting results?

Answer: We confirmed the function of the SLC transporters in HEK293 or Flp293 cells by using p-14C-aminohippuric acid and methyl-3H-4-phenylpyridinium that are the substrates of each transporter. We added the figure legends in this study.

3.Data of Fig.2 are presented as mean and standard deviation (SD), please provide the sample number (n=?)

Answer: Thanks. We conducted the experiments with a sample size of n=4. We added it in the materials and methods.

4.Why did the results of thyroid gland, stomach, small intestine, and urinary bladder show as % ID, the other organs (blood, heart, liver and kidney) show as % ID/g?

Answer: Since 125I-AP usually accumulates in cells and tissues, we can calculate radioactive accumulation by using %ID/g corrected with the weight of the tissues. However, the thyroid gland of mice is quite small and the stomach and small intestine can’t completely remove the contents. So, it is difficult to measure the accurate weight of these tissues and organs. Therefore, these tissues and organs were shown in %ID. In addition, urine accumulates into cavity of the urinary bladder. So, it is undesirable that the radioactivity of urine is corrected by the weight of the urinary bladder. Thus, the urinary bladder was also shown in %ID. These are general units in research fields of nuclear medicine imaging.

5.Please supply the procedure after inhibitor incubated in section 2.4, such as inhibitor incubated time, detected time, washed procedure, protein extracting procedure and so on.

Answer: We added experimental information in materials and methods. Incubation time of both inhibitor and 125I-AP was 5 min. After incubation, the cells were washed twice with 600 µL PBS and lysed with 500 µL of 0.1 M NaOH. Radioactivity was measured using an auto well gamma counter. As for protein extraction, we exposed intracellular proteins by disrupting the cell membrane with NaOH, and measured the amount of proteins in the lysate using bicinchoninic acid protein assay kit (Thermo Fisher Scientific).

6.Please supply analysis of time-activity curves for important organs.

Answer: The graph of time-activity curves was not showed in this manuscript because the biological distribution (Table 1) had already showed time-activity curves after 125I-AP administration.

7.Why did the author not provide the experiment data about the disease or related transporters deficient mouse models after the oral administration of 125I-AP?

Answer: We have already written it as limitation in this Discussion. It is not easy to make OATP- or OAT-deficient mouse models because technique is needed to modify mouse genes. In addition, there is no disease mouse models which decreased expression of OATP and/or OAT. Therefore, we used a mouse model with inhibitors for these transporters to inhibit gastrointestinal absorption in the small intestine.

Round 2

Reviewer 1 Report

Authors have reasonably answered all my comments.

Author Response

Thank you for your reports. We appreciate it.